# General practitioner visits after SARS-CoV-2 omicron compared with the delta variant in children in Norway: a prospective nationwide registry study

Sigurd Storehaug Arntzen,[1] Hege Marie Gjefsen,[1] Kjetil Elias Telle,[1] Karin Magnusson,[1] Ketil Størdal,[2] Siri Eldevik Håberg,[3] Jonas Minet Kinge [1,3,4]

[1]Cluster for Health Services Research, Norwegian Institute of Public Health, Oslo, Norway
[2]Division of Paediatric and Adolescent Medicine, University of Oslo, Oslo, Norway
[3]Centre for Fertility and Health, Norwegian Institute of Public Health, Oslo, Norway
[4]Department of Health Management and Health Economics, University of Oslo, Oslo, Norway

**Correspondence to**
Dr Jonas Minet Kinge; Jonas. minet.kinge@fhi.no

## ABSTRACT

**Background** SARS-CoV-2 infection in children is followed by an immediate increase in primary care utilisation. The difference in utilisation following infection with the delta and omicron virus variants is unknown.

**Objectives** To study whether general practitioner (GP) contacts were different in children infected with the omicron versus delta variant for up to 4 weeks after the week testing positive.

**Setting** Primary care.

**Participants** All residents in Norway aged 0–10. After excluding 47 683 children with a positive test where the virus variant was not identified as delta or omicron and 474 children who were vaccinated, the primary study population consisted of 613 448 children.

**Main outcome measures** GP visits.

**Methods** We estimated the difference in the weekly share visiting the GP after being infected with the delta or omicron variant to those in the study population who were either not tested or who tested negative using an event study design, controlling for calendar week of consultation, municipality fixed effects and sociodemographic factors in multivariate logistic regressions.

**Results** Compared with preinfection, increased GP utilisation was found for children 1 and 2 weeks after testing positive for the omicron variant, with an OR of 6.7 (SE: 0.69) in the first week and 5.5 (0.72) in the second week. This increase was more pronounced for children with the delta variant, with an OR of 8.2 (0.52) in the first week and 7.1 (0.93) in the second week. After 2 weeks, the GP utilisation returned to preinfection levels.

**Conclusion** The omicron variant appears to have resulted in less primary healthcare interactions per infected child compared with the delta variant.

## WHAT IS ALREADY KNOWN ON THIS TOPIC

⇒ SARS-CoV-2 in children is known to lead to an immediate increase in primary care utilisation for 1–2 weeks after a positive test, before quickly falling back to preinfection utilisation levels. The difference in primary care utilisation following infection with the delta and omicron virus variants is unknown.

## WHAT THIS STUDY ADDS

⇒ A sudden increase in primary care use was observed in children who tested positive for SARS-CoV-2 in the 2 weeks after the test. After the first 2 weeks, the primary care utilisation returned to preinfection levels.

⇒ The increase in primary care utilisation was higher for children infected with the delta than the omicron variant.

## HOW THIS STUDY MIGHT AFFECT RESEARCH, PRACTICE OR POLICY

⇒ Compared with the delta variant, the omicron variant is likely to result in less pressure on primary care services per infected child.

## INTRODUCTION

SARS-CoV-2 for children is often referred to as leading to milder symptoms than in adults, and recent studies found no increase in specialist care in children following infection.[1] [2] However, SARS-CoV-2 infection in children is followed by an immediate increase in primary care utilisation, and reports have discussed whether the omicron variant might cause more severe symptoms than the delta variant in children.[2–4]

Survey data can be used to determine patterns of healthcare need following infection; however, reporting and response bias may affect the accuracy of the estimates. Except for a few studies using registry data from a period dominated by delta and other earlier variants, little is known about the impact of SARS-CoV-2 on post-COVID healthcare utilisation in children.[2] Furthermore, we also do not know whether healthcare use among children and adolescents increases after initial omicron infection and whether this increase, if any, is comparable to the increase in utilisation after infection with the delta variant. Such knowledge could be used to upscale or downscale the healthcare services.

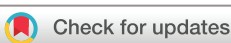

Because the omicron variant has been found to cause less severe symptoms than the delta variant in adults, we hypothesised that a comparable pattern would be found for children. The aim of this analysis was to compare general practitioner (GP) contacts among children in the 4 weeks after being infected with the omicron or delta variant.

## METHODS

Data used for this project were from the emergency preparedness register (Beredt C19). The establishment of an emergency preparedness register forms part of the legally mandated responsibilities of the Norwegian Institute of Public Health during epidemics.

### Data sources

To estimate the share of children aged 0–10 years visiting the GP after being infected with delta or omicron variant we used population-wide longitudinal registry data from Norway. Beredt C19 is an emergency preparedness register that aims to rapidly provide ongoing overview and knowledge of the prevalence, causal relationships and consequences of the COVID-19 epidemic in Norway. It includes information from various data sources that are updated daily, including the Norwegian Surveillance System for Communicable Diseases (all testing and screening for SARS-CoV-2), the National Population Register (age, sex, country of birth), the National Immunization Register (vaccination status) and the Norway

Control and Payment of Health Reimbursement (all physical and electronic consultations with all GPs). A more in-depth summary of the data sources used for our analysis is available in the online supplemental table 1.

### Study population

We followed all Norwegian residents aged 0–10 years from 29 November 2021 to 23 January 2022. Figure 1 shows the share of the sequenced PCR tests that were delta or omicron variant from 29 November to 23 January. Children who tested positive but whose tests were not sequenced, and children who had been vaccinated, were excluded from the analysis. The upper age cut-off at 10 was set as children who turned 11 at the start of the period, turned 12 during the study period and thus become eligible for the vaccine.

The categorical outcome variable for GP contact was set to 1 if the individual had at least one physical consultation or e-consultation with a GP in a week, and 0 otherwise. In Norway, consultations with the GP are free for children aged below 16. The GPs serve as the first line in the healthcare services, prescribing medicines and performing simple procedures, and referring patients to further care when necessary.

### Statistical design

We constructed a data set including one observation per individual per week from week 48 of 2021 until the third week of 2022. Each week, each individual could either be registered with a GP consultation or not. For

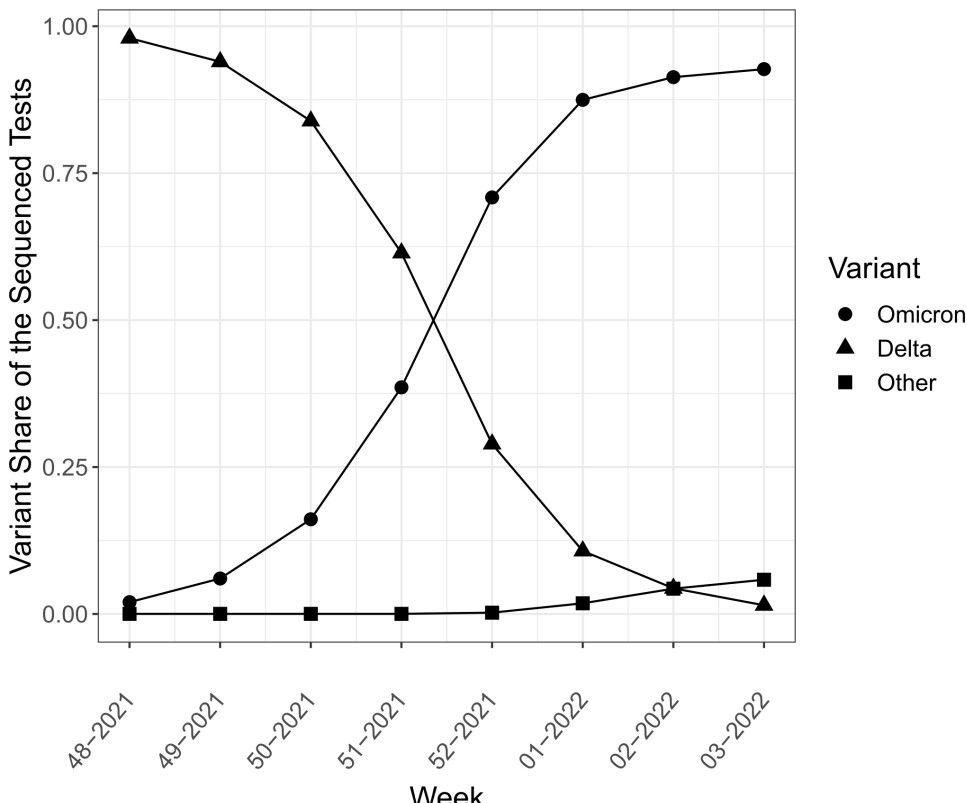

**Figure 1** The development of delta and omicron cases in the estimated sample, week 48 (2021) to week 3 (2022). Share of sequenced samples with confirmed delta, omicron and other results.

**Table 1** Summary statistics of the estimation sample

|  | Omicron | Delta | Rest |
|---|---|---|---|
| n | 7046 | 14369 | 592033 |
| Person-weeks | 56368 | 114952 | 4736264 |
| Age, mean (SD) | 6.1 (3.1) | 6.9 (2.8) | 5.1 (3.2) |
| Born in foreign country (%) | 7.2 | 6.3 | 5.5 |
| Parents born in foreign country (%) | 53.7 | 43.0 | 38.8 |
| Boys (%) | 51.3 | 51.4 | 51.3 |

the individuals who were infected with COVID-19, we constructed an index week of infection. For each index week, persons with omicron or delta were compared with persons without omicron or delta. Event time was indicated relative to the index week of COVID-19 infection for each person and was our main variable of interest explaining primary healthcare use for omicron and delta variants, respectively.

Multivariate logistic regression was used to estimate adjusted ORs (aOR) with 95% CIs for GP consultations. Exploiting the longitudinal nature of the data, we used an event study design,[5 6] controlling for calendar week of consultation, municipality fixed effects and sociodemographic characteristics.

The event study is especially well suited when the timing of events varies across groups in the population, there is a high number of units not experiencing an event and any measured association might vary over time.[7] The approach is widely used in social sciences and now also increasingly popular in epidemiology and public health as it can display an abnormal shift in trend, and attribute that shift to an event.[8–12]

The temporal aORs of being infected by omicron and delta were estimated from 5 weeks before to 4 weeks

after the week 0 of infection. We regress weeks to and from confirmed positive test on binary GP visits using the following expression:

$$\sigma\left(y_{iw}\right) = 1/\left(1 + e^{-y_{iw}}\right)$$

$$y_{iw} = \theta_w + \theta_i + \sum_{k=-5,k\neq-1}^{k=5} \delta_{k(iw)}\alpha_k + \sum_{k=-5,k\neq-1}^{k=5} \sigma_{k(iw)}\beta_k$$

$y_{iw}$ is the outcome for individual $i$ in week $w$, that is, GP visits. $\theta_w$ is a set of dummy variables for calendar week accounting for any changes in the inclination to visit a GP due to, for example, capacity constraints or holidays. $\theta_i$ denotes background characteristics for individual $i$ including gender, age dummies, parental country background and the child's country background, as well as municipality fixed effects. $\delta_k$ is a set of time dummy variables, indicating the event time, that is, the number of weeks $k$ relative to the week in which the individual got infected with the delta virus, taking the value 0 if not being infected with the delta virus. Similarly, $\sigma_k$ is a set of dummy variables for event time in the case of infection with the omicron variant. The week prior to the infection, k=−1, is used as our reference value, and this value is therefore omitted from the regression. Our primary parameters of interests were the $\beta_k$ and $\alpha_k$ attached to the event time dummies. These captured the changes in the probability of visiting the GP among the children infected with delta and omicron relative to the comparison group consisting of uninfected or non-tested. $\epsilon_{iw}$ was the SE clustered at the municipality level.

The coefficients $\delta_k$ and $\sigma_k$ for k<0 indicated the GP use develops prior to infection time, while k>0 described how the outcome changed after getting infected with either delta or omicron virus. Hence, the event study framework allows for testing whether infected children followed the same patterns for GP visits as the non-infected prior to infection, and whether this pattern changed after the week 0 of infection. A discontinuous jump in the

**Table 2** Descriptive statistics of the estimated sample

| | Omicron | | | Delta | | |
|---|---|---|---|---|---|---|
| Relative week | Person-weeks with GP appointment | Persons infected | Share with GP appointment | Person-weeks with GP appointment | Persons infected | Fraction with GP appointment |
| −5 | 145 | 4975 | 0.03 | 19 | 692 | 0.03 |
| −4 | 176 | 6126 | 0.03 | 55 | 1622 | 0.03 |
| −3 | 147 | 6600 | 0.02 | 113 | 3452 | 0.03 |
| −2 | 120 | 6902 | 0.02 | 187 | 7445 | 0.03 |
| −1 | 125 | 7024 | 0.02 | 259 | 11636 | 0.02 |
| 0 | 846 | 7046 | 0.12 | 2285 | 14369 | 0.16 |
| 1 | 674 | 6263 | 0.11 | 1795 | 14351 | 0.13 |
| 2 | 136 | 4133 | 0.03 | 335 | 14177 | 0.02 |
| 3 | 60 | 2071 | 0.03 | 282 | 13677 | 0.02 |
| 4 | 17 | 920 | 0.02 | 304 | 12747 | 0.02 |

GP, general practitioner.

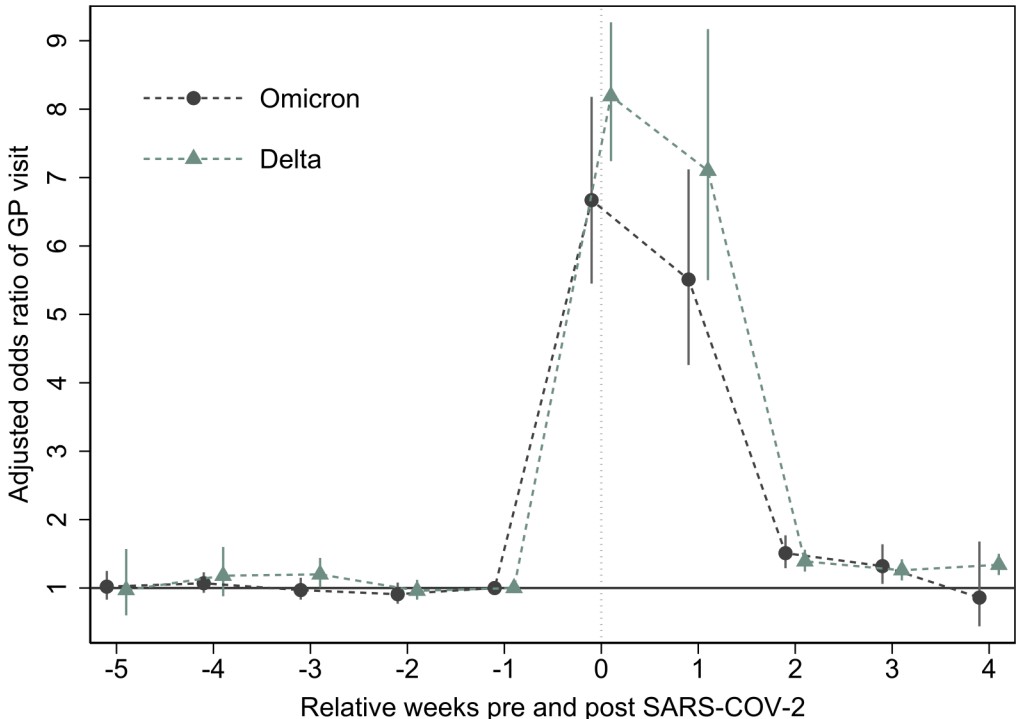

**Figure 2** Event study analysis of pre-SARS-CoV-2 and post-SARS-CoV-2 trends in ORs for GP utilisation by variant. Estimates from logistic regression analysis on weekly data are reported. Adjusted ORs were estimated for indicator variables for relative weeks to omicron and delta and the omitted category defined as 1 week prior to infection, that is, week 1. ORs were adjusted for age by including age indicators for 1 year age group, sex, indicator for birth country and parental birth country and municipality of residence. SEs were clustered at municipality level. GP, general practitioner.

probability of visiting the GP around week 0 indicates an estimated difference in the probability of visiting the GP between the individuals infected with omicron or delta, and our comparison groups consisting of non-infected individuals.

In the online supplemental appendix, we present robustness checks of our results by varying the time period of our analysis, age-stratified analysis and results of analysis with additional adjustment for municipality-specific time trends.

### Patient and public involvement

No patients were involved in setting the research question or the outcome measures, nor were they involved in developing plans for recruitment, design or implementation of the study. No patients were asked to advise on interpretation or writing up of results. There are no plans to disseminate the results of the research to study participants or the relevant patient community.

### RESULTS

In total, 661 587 children aged 0–10 years were residing in Norway in the study period. After excluding 474 children who were vaccinated and 47 665 children with positive tests during the study period where the virus variant was not identified as delta or omicron, the primary study population consisted of 613 448 children comprising 4 907 584 person-weeks.

Figure 1 shows that delta was the dominant virus variant at the beginning of our study period, while omicron was the dominant variant at the end of our study period. Persons with omicron were older and more often born abroad than persons with delta, though the sex distribution was similar (table 1).

Table 2 shows the descriptive statistics of the estimated sample, indicating that the fraction that visited the GP in the weeks before infection was similar for the children later infected with omicron and delta, and for those who either were not infected or not tested for SARS-CoV-2.

The event study plot shows higher GP utilisation following omicron, compared with delta (figure 2). The aOR of 8.19 (SE: 0.52) in the first week and aOR 7.10 (0.93) in the second week after delta were higher than comparable estimates for omicron of 6.67 (0.69) and 5.51 (0.72) in weeks 1 and 2, respectively. Higher GP utilisation was also found for children 4 weeks after testing positive for both omicron and delta. Higher utilisation was also found 4 weeks after testing positive for delta but returned to preinfection levels in week 4 for omicron cases (see table 3 for details).

Results from when varying included time period were similar (online supplemental table 2). The results were also replicated in each age stratum (online supplemental tables 3A–C). The results were also robust to analysis including municipality-specific time trends (online supplemental table 4).

**Table 3** Event study estimates of the effect of relative week according to week delta and omicron variant on GP use from 29 November 2021 to 23 January 2022

| Relative week | OR | SE | P value | Lower CI | Upper CI | Test for equal OR (P value)* |
|---|---|---|---|---|---|---|
| Omicron | | | | | | |
| Week −5 | 1.02 | 0.11 | 0.86 | 0.83 | 1.25 | 0.85 |
| Week −4 | 1.07 | 0.08 | 0.37 | 0.93 | 1.23 | 0.53 |
| Week −3 | 0.97 | 0.08 | 0.75 | 0.83 | 1.15 | 0.13 |
| Week −2 | 0.91 | 0.08 | 0.27 | 0.77 | 1.08 | 0.62 |
| Week −1 | 1 | | | | | |
| Week 0 | 6.67 | 0.69 | 0.00 | 5.45 | 8.18 | 0.01 |
| Week 1 | 5.51 | 0.72 | 0.00 | 4.26 | 7.12 | <0.01 |
| Week 2 | 1.51 | 0.12 | 0.00 | 1.29 | 1.77 | 0.46 |
| Week 3 | 1.32 | 0.15 | 0.01 | 1.06 | 1.64 | 0.70 |
| Week 4 | 0.86 | 0.29 | 0.65 | 0.44 | 1.68 | 0.18 |
| Delta | | | | | | |
| Week −5 | 0.97 | 0.24 | 0.90 | 0.60 | 1.57 | |
| Week −4 | 1.18 | 0.18 | 0.27 | 0.88 | 1.60 | |
| Week −3 | 1.20 | 0.11 | 0.04 | 1.01 | 1.44 | |
| Week −2 | 0.96 | 0.07 | 0.63 | 0.83 | 1.12 | |
| Week −1 | 1 | | | | 1 | |
| Week 0 | 8.19 | 0.52 | 0.00 | 7.24 | 9.27 | |
| Week 1 | 7.10 | 0.93 | 0.00 | 5.50 | 9.17 | |
| Week 2 | 1.39 | 0.08 | 0.00 | 1.24 | 1.56 | |
| Week 3 | 1.26 | 0.08 | 0.00 | 1.11 | 1.42 | |
| Week 4 | 1.34 | 0.08 | 0.00 | 1.19 | 1.50 | |

Regression results from the main specification using a logit model. Number of person-weeks=4 906 952. We included controls for year of birth, calendar week, region of residence, country of birth, sex and parents' country of birth. SEs were clustered at the municipality level.
*The column shows p values from Wald tests of equal OR for omicron versus delta, based on the regression output.
GP, general practitioner.

## DISCUSSION

In this study of 613 448 Norwegian children aged 0–10 years, we found increased GP utilisation for children 1 and 2 weeks after testing positive for the omicron variant, with similar and more pronounced increases for children with the delta variant. Our findings suggest that omicron will place less pressure on the primary care services per case.

In the week following positive test 16% of children with delta visited the GP, compared with 12% with omicron. After 1 week, this per cent dropped to 13% and 11% for children with delta and omicron, respectively. This suggests less pressure on the services from omicron. However, if case load is substantially increased, it might outweigh the reduced pressure per case. The overall pressure on the healthcare system is a product of how many are infected and their inclination to use the healthcare system. Even if the inclination is overall lower with omicron, the pressure on the healthcare system can be higher if the number of infected is sufficiently high.

A number of reports of the omicron variant in adults suggested less serious illness already a few weeks after the initial reports of the new variant.[13] Thus, our findings may be affected by parents' perceived less need to seek healthcare after omicron in their children than after delta. However, a key strength of our analysis was that both delta and omicron were circulating during our study period, and the parents were not informed which strain they were infected with. Hence, given that we adjust for overall time trend, it is unlikely that a difference in the perceived relative severity of the two strains was the main driver of our results. One exception is that if infection with delta and omicron variants differed in symptoms, known to the parents, the parents could have reacted accordingly.

There might very well have been fluctuations in the inclination for GP visits over time. We adjusted for calendar week in our regression models, making it unlikely that the results were driven by general changes in the inclination to contact the GP. As both virus strains were present

in all weeks included in our analysis, changes due to, for example, high pressure on GPs or an overall impression of omicron being less severe should be addressed by the week fixed effects. In addition, our analysis included data covering all children residing in Norway, which reduced attrition and sample selection bias.

Based on national recommendations in Norway, healthy children below age 12 are mostly unvaccinated.[14] PCR tests are freely available and a proportion of these are screened for variants of concern. Several of the most populous municipalities in Norway had implemented mass testing in the schools before the first case of omicron was detected in Norway in late November 2021. However, throughout the study period, it was the norm that all positive rapid antigen home tests had to be confirmed with a PCR test. The increasing use of rapid antigen home tests was therefore unlikely to seriously bias our estimates.

A limitation of the study was that asymptomatic children might have been less likely to have a PCR test. If, for example, asymptomatic SARS-CoV-2 infection was more common for the omicron than the delta variant, we might have underestimated the difference between them. Also, our findings may not be generalisable to countries without equal and free access to healthcare and PCR testing for SARS-CoV-2 for all inhabitants. Finally, there have been variation in the tests that were sequenced over time. For capacity reasons, samples suspected to contain the omicron variant were prioritised for variant analyses for a part of the period. This might have led to variation in the tests that were sequenced over time and therefore potentially the composition of the groups.

## CONCLUSION

Our findings showed that per positive test in children aged 0–10 years, the omicron variant was likely to result in fewer consultations per positive tested children than the delta variant. However, the omicron variant was still associated with higher total number of consultations, and could lead to a high burden on the healthcare system when the number of children infected with omicron is high.

**Contributors** SSA designed the study, had full access to all data in the study, performed the data management, interpreted the results and drafted the article. HMG designed the study, interpreted the results and drafted the article. KET, KM, KS and SEH conceptualised and critically revised the study for intellectual content, interpreted the results and drafted the article. JMK designed the study, had full access to all data in the study, performed the statistical analyses, interpreted the results and drafted the article. JMK is responsible for the overall content as guarantor

**Funding** The study was internally funded by the Norwegian Institute of Public Health and externally by the Research Council of Norway (project number: 262700).

**Disclaimer** The funding sources had no influence over the design and conduct of the study; collection, management, analysis and interpretation of the data; preparation, review or approval of the manuscript; or decision to submit the manuscript for publication.

**Competing interests** None declared.

**Patient and public involvement** Patients and/or the public were not involved in the design, or conduct, or reporting, or dissemination plans of this research.

**Patient consent for publication** Not applicable.

**Ethics approval** The Ethics Committee of South-East Norway confirmed (4 June 2020; 153204) that external ethical board review was not required.

**Provenance and peer review** Not commissioned; externally peer reviewed.

**Data availability statement** Data may be obtained from a third party and are not publicly available.

**ORCID iD**
Jonas Minet Kinge http://orcid.org/0000-0002-2789-6113

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
