## [Reviewer comments · BMJ Paediatrics Open]

This paper was submitted to a another journal from Archives of Disease in Childhood but declined for publication following peer review. The authors addressed the reviewers' comments and submitted the revised paper to BMJ Paediatrics Open. The paper was subsequently accepted for publication at BMJ Paediatrics Open.

ARTICLE DETAILS

TITLE (PROVISIONAL)	General practitioner visits after SARS-CoV-2 omicron compared to the delta variant in children in Norway: a prospective nationwide registry study
AUTHORS	Arntzen, Sigurd Storehaug Gjefsen, Hege Marie Telle, Kjetil Elias Magnusson, Karin Stordal, Ketil Håberg, Siri Eldevik Kinge, Jonas

VERSION 1 – REVIEW

REVIEWER	Reviewer name: Dr. Andrew Riordan Institution and Country: Alder Hey Childrens NHS Foundation Trus, Ireland Competing interests: None
REVIEW RETURNED	21-Mar-2022

GENERAL COMMENTS	This letter takes Primary care data from the 29/11/21 until 23/1/22 for 613,448 Norwegian children aged 10 yrs and under to assess the risk of these children being seen by a GP in the 5 weeks before and after testing positive for either Delta or Omicron COVID. The adjusted Odds Ratio in the first and second week after positive tests were higher for delta than for omicron. This is an interesting finding. The letter needs to be brief so it's hard to assess the analysis from the information provided (although the supplementary file gives more data). The actual numbers of GP consultations and the robustness of the analysis is not clear. Might this be better as a short report, to allow fuller presentation of the data and statistical review?
---

Version 2 – Author's Response

Dear Prof. Imti Choonara, Thank you for the opportunity to resubmit an improved version of our manuscript (bmjpo-2022- 001502). The editor requested an original article, and the reviewer requested a short report to allow fuller presentation of the data. Hence, we now submit a thoroughly revised full original article for your consideration. Responses to each comment are marked in grey below. We hope that the changes made meet your expectations and look forward to your consideration. Sincerely, Jonas Minet Kinge 08-Apr-2022 bmjpo-2022-001502 - "General practitioner visits after SARS-CoV-2 Omicron compared to the Delta variant in children: a prospective nationwide registry study" Dear Dr. Kinge, Following review of your article to BMJ Paediatrics Open,

we invite you to submit a major revision. The review comments can be found at the end of this email, together with any comments from the Editorial Office regarding formatting changes or additional information required to meet the journal's policies at this time. Please note that your revision may be subject to further review and that this initial decision does not guarantee acceptance at this time. To submit your revised article please click this link: *** PLEASE NOTE: This is a two-step process. After clicking on the link, you will be directed to a webpage to confirm. *** https://mc.manuscriptcentral.com/bmjpo?URL_MASK=cd073f41bdf6453b9738159b71599009. Alternatively, you can log on to your Author Dashboard in ScholarOne and under "Action" click "create a revision". Please read and respond to all of the peer review comments. You should provide a point-by-point response to explain any changes you have (or have not) made to the original article and be as specific as possible in your responses. The original files will be available to you when you start your revision. Please delete any files that you intend to replace with updated versions and upload the following using the appropriate file designation: "Main Document" - This is a clean copy (without tracked or highlighted changes) of your revised article. Please delete your original submission file. "Main Document - marked copy" - This is the edited version of your original article, including edits to address the peer review comments. Any changes have been highlighted using a track change function or bold or coloured text. Please replace any other files that have been updated e.g. Images, forms The reviewers' comments, your response, and the previous versions of your article will be published as supplementary information alongside the final version of your article. Information relating to your article, including author names and affiliations, title, abstract and required statements (e.g. competing interests, contributorship, funding) will be taken directly from the information held in ScholarOne, and not from the article file. Please check that this information has been entered correctly and has been updated as appropriate. If your revised article is accepted, you will only be able to make minor changes (e.g. correction of typesetting errors and proof stage) prior to publication. Please submit your revised article by 08-May-2022. If we have not received it by this date, the opportunity to submit a revision will expire and your article may be treated as a new submission. If you need to request an extension, please contact the Editorial Office as soon as possible. Thank you for submitting your article to BMJ Paediatrics Open; we look forward to receiving your revision. If you have any queries, please contact the Editorial Office at info.bmjpo@bmj.com. Kind regards, Associate Editor, BMJ Paediatrics Open Prof. Imti Choonara Editor in Chief, BMJ Paediatrics Open

Formatting Amendments (where applicable): Editor(s)'

Comments to Author (if any): Your article would be better as an original article. There is no minimum word count for an original article. Your supplementary data needs to be in the main paper. Please note that the revision will go out for peer review (including stats review) We thank the editor for allowing us to resubmit an original article. We have thoroughly revised the manuscript and expanded each section, especially the methods section. We have included the supplementary data in the main paper. Reviewer: 1 Comments to the Author This letter takes Primary care data from the 29/11/21 until 23/1/22 for 613,448 Norwegian children aged 10 yrs and under to assess the risk of these children being seen by a GP in the 5 weeks before and after testing positive for either Delta or Omicron COVID. The adjusted Odds Ratio in the first and second week after positive tests were higher for delta than for omicron. This is an interesting finding. The letter needs to be brief so it's hard to assess the analysis from the information provided (although the supplementary file gives more data). The actual numbers of GP consultations and the robustness of the analysis is not clear. Might this be better as a short report, to allow fuller presentation of the data and statistical review? We thank the reviewer for commenting on our manuscript. We have expanded the manuscript text,

including a more detailed explanation of the methods and included the supplementary data in the manuscript.

VERSION 3 – REVIEW

REVIEWER	Reviewer name: Dr. Peter Flom Institution and Country: Peter Flom Consulting, United States Competing interests: None
REVIEW RETURNED	20-May-2022
GENERAL COMMENTS	I confine my remarks to statistical aspects of this paper. I think the general approach is OK (but see my first comment, below) but I have some questions and issues to resolve before I can recommend publication. General I traced the citations of the general model back a couple of generations. It seems to be OK, in general, although it is a rather obscure model (as far as I could tell). Why did the authors choose this model instead of a nonlinear multilevel model? p. 7 The explanation of the terms in the formula would be easier to follow if each term was taken in turn. That is, first Y_{iw}, then θ_w then θ_i and so on. Also, α_k and β_k are not explained. And, I am not sure I understand what δ_k is. "a set of time variables" isn't clear to me. What sort of thing are these? Where is the logit or the exponentiation? The model, as written, is linear regression, not logistic. Figure 1 Stacked bar charts are not a great method (see the work of William Cleveland), and this particular figure has some particular problems: The label on the y axis is overwritten (although this may be because of formatting imposed by BMJ - it seems to be line num, the background grid makes it harder to read the actual bars, and omikron is a typo. I would make a line graph, with "week" on the x axis, "proportion" on the y axis and a line for omicron, delta, and other. (3 lines). Figure 2 I would jitter the time, so that the standard error bars don't overlap Peter Flom

REVIEWER	Reviewer name: Dr. Patrick Aldridge Institution and Country: Frimley Park Hospital NHS Foundation Trust, United Kingdom of Great Britain and Northern Ireland Competing interests: None
REVIEW RETURNED	15-Jun-2022
GENERAL COMMENTS	Thanks. Overall - interesting read, written in wrong tense. Needs to be rewritten. Will add to literature so think should be eventually published as will be interesting to other countries. Paper suggests less GP attendances with Omicron & regression analysis supporting this. The odds ratios don't appear that dissimilar to draw the authors conclusions. I am not convinced from this paper that Omicron infected children had less attendances compared to delta. Omicron was more prevalent at the end of study period (authors note this) and parental behaviour is not adequately considered/addressed.

	I am not a statistician so will not dwell too much on calculations used but the odds ratio is calculated from groups with different numbers of infected children (7,000 vs 14,000 roughly) so not sure you can draw conclusions about higher odds ratios (delta vs Omicron). I think this article is written in the wrong tense throughout (present rather than past) and needs adjusting, for example in abstract conclusion 'The omicron variant is likely to result in less pressure on primary health care services for children, compared with the delta variant'. Suggest 'In children aged 0-10yrs the omicron variant appears to have led to reduced primary health care interactions, compared to the delta variant' Abstract - ' SARS-CoV-2 in children is known to lead to an immediate increase in primary care utilization for 1-2 weeks after a positive test' - there is no reference to where this fact comes from in the introduction & is quite a bold statement. Think needs to be toned down/removed unless it can be supported in literature. Sample - Why age 0-10 yrs only? You mention up to 12 yrs not immunised so why not include up to 12yrs? Some analysis on age groups may prove useful but given word limit may not be possible. Where there any age groups who attended GP more i.e was there a disproportionate increase in any age group? We know babies attended UK hospitals more (for example https://bmjpaedsopen.bmj.com/content/6/1/e001345#DC2) my own work, just an example. Page 9 'Our findings suggest that omicron will place less pressure on the primary care services per case. However, given the higher transmissibility of the omicron than the delta variant it can still lead to a high burden on the health care system' - this is confusing. You allude to lower GP attendances with Omicron in abstract/conclusion but given higher transmittance you argue it probably won't reduce attendances. Think need to adjust conclusions in light of this statement. Page 10-11 'Finally, for continuous surveillance purposes, 25% of SARS-CoV-2 positive samples or up to 100 samples per week per local laboratory is sent to a reference laboratory for whole genome sequencing. When omicron emerged in Norway in late November 2021, the laboratories were requested to... this whole section & including 4-5 sentences afterwards are too long and should be shortened. Tables 1 & 2 - confusing as Omicron & delta in 1st/2nd column in Table 1 & 2nd/1st column in Table 2. Better alignment for reader would be helpful.
--	--

Version 3 – Author’s Response

Dear Dr. Malcolm Brodli & Prof. Imti Choonara, Thank you for the opportunity to resubmit an improved version of our manuscript (bmjpo-2022- 001502.R1). We would like to thank the editorial board and the expert reviewers for a careful review and consideration of our study. Please see the detailed point-to-point responses and actions to the editors’ and the reviewers’ comments beneath. Our page references refer to the revised version of the manuscript, without marked changes. Responses to each comment are marked in grey below. We look forward to your consideration of our revised manuscript. Sincerely, Jonas Minet Kinge

VERSION 4 – REVIEW

REVIEWER	Reviewer name: Dr. Patrick Aldridge Institution and Country: Frimley Park Hospital NHS Foundation Trust, United Kingdom of Great Britain and Northern Ireland Competing interests: None
REVIEW RETURNED	13-Jul-2022
GENERAL COMMENTS	Thank you for addressing the points I had raised and I have no more concerns.